# Changes in Organic Acids, Phenolic Compounds, and Antioxidant Activities of Lemon Juice Fermented by *Issatchenkia terricola*

**DOI:** 10.3390/molecules26216712

**Published:** 2021-11-05

**Authors:** Biao Liu, Dongxia Yuan, Qiaoyue Li, Xin Zhou, Hao Wu, Yihong Bao, Hongyun Lu, Ting Luo, Jinling Wang

**Affiliations:** 1School of Forestry, Northeast Forestry University, No. 26, Hexing St., Harbin 150040, China; 1311153858@nefu.edu.cn (B.L.); dongxia15589577379@163.com (D.Y.); li2352426450@163.com (Q.L.); zx19855755122@163.com (X.Z.); wh13796652959@163.com (H.W.); zxc943855156@sina.com (Y.B.); 2Key Laboratory of Forest Food Resources Utilization of Heilongjiang Province, No. 26, Hexing St., Harbin 150040, China; 3Department of Food Science and Nutrition, Zhejiang University, Hangzhou 310058, China; luhongyun@zju.edu.cn; 4State Key Laboratory of Food Science and Technology, Nanchang University, No. 999, Xuefu St., Nanchang 330047, China; ting.luo@ncu.edu.cn

**Keywords:** *Issatchenkia terricola*, deacidification, phenolic compounds, organic acid, antioxidant activity, principal component analysis (PCA)

## Abstract

High content of citric acid in lemon juice leads to poor sensory experience. The study aimed to investigate the dynamics changes in organic acids, phenolic compounds, and antioxidant activities of lemon juice fermented with *Issatchenkia terricola* WJL-G4. The sensory evaluation of fermented lemon juice was conducted as well. *Issatchenkia terricola* WJL-G4 exhibited a potent capability of reducing the contents of citric acid (from 51.46 ± 0.11 g/L to 8.09 ± 0.05 g/L within 60 h fermentation) and increasing total phenolic level, flavonoid contents, and antioxidant activities compared to those of unfermented lemon juice. A total of 20 bioactive substances, including 10 phenolic acids and 10 flavonoid compounds, were detected both in fermented and unfermented lemon juice. The lemon juice fermented for 48 h had better sensory characteristics. Our findings demonstrated that lemon juice fermented with *Issatchenkia terricola* exhibited reduced citric acid contents, increased levels of health-promoting phenolic compounds, and enhanced antioxidant activities.

## 1. Introduction

Lemons, a small evergreen perennial tree of the genus *Citrus* in the family Rutaceae, are mainly distributed in tropical and subtropical regions such as Argentina, Brazil, China, India, Australia, and the United States. According to the Food and Agriculture Organization of the United Nations (FAO) [1], the total annual production of lemons and limes in China reached 2.330 million tons, ranking third in the world. China has a long history of lemon cultivation, which is mainly distributed in Sichuan, Chongqing, Yunnan, Guangdong, Guangxi, Fujian, Hainan, and Taiwan (www.atlasbig.com “World’s top Lemon producing Countries”, accessed on 27 October 2021).

Lemon fruits are rich in nutrients and bioactive compounds [2] such as vitamins, dietary fiber, minerals, carotenoids, phenolic compounds, and essential oils [3]. Among them, phenolic compounds are widely reported to exert antioxidant [4], anti-inflammatory [5], and antibacterial activities [6], as well as prevent cancer and cardiovascular diseases [7]. Lemon juice is one of the major processed products of lemon fruits [8]. However, the high content of citric acid in lemon juice, which results in poor sensory experience, limits its application in the food industry. In our previous work, the yeast with the ability to degrade 20 g/L of citric acid was found in red raspberry fruits and named *Issatchenkia terricola* WJL-G4 [9]. Fermentation raspberry juice with *Issatchenkia terricola* WJL-G4 had higher total flavonoid contents and higher acceptance compared to that of unfermented raspberry juice [10].

The main organic acid in lemon juice is citric acid, which is about 73.94 g/L and accounts for more than 90% of the total acids in lemon juice [11]. Studies on acid reduction in lemon juice have not been reported. Therefore, we supposed that fermentation with *Issatchenkia terricola* WJL-G4 could reduce the citric acid in lemon juice and might improve the acceptance of lemon juice. The study aimed to investigate the dynamics changes in organic acids, phenolic compounds, and antioxidant activities during 60 h fermentation of lemon juice fermented with *Issatchenkia terricola* WJL-G4 and to evaluate the acceptance of lemon juice after fermentation. This paper presents for the first time to ferment lemon juice with *Issatchenkia terricola* WJL-G4 to reduce the content of citric acid while increasing the health-promoting potential by increasing the levels of phenolic compounds and antioxidant activities.

## 2. Results

### 2.1. Physicochemical Properties and Organic Acids of Fermented and Control Lemon Juice

The physicochemical characteristics of the fermented and control lemon juice during different fermentation processes are shown in Table 1. During fermentation with *Issatchenkia terricola* WJL-G4, the contents of total sugar decreased from 27.97 ± 3.28 g/L to 6.56 ± 0.21 g/L; the contents of reducing sugar decreased from 11.69 ± 0.70 g/L to 1.10 ± 0.02 g /L; the contents of total titratable acidity decreased from 58.59 ± 0.39 g/L to 12.20 ± 0.04 g/L; and the variation of pH ranged from 2.61 ± 0.01 to 4.31 ± 0.01. In the control group, contents of total sugar showed a floating change from 27.97 ± 3.28 g/L to 21.00 ± 0.00 g/L; contents of reducing sugar showed a slight increase from 11.69 ± 0.70 g/L to 13.18 ± 0.94 g/L; contents of total titratable acidity did not show significant changes; and pH value showed a slight increase, from 2.61 ± 0.01 to 2.73 ± 0.01.

The contents of organic acids, including citric acid, malic acid, and succinic acid, the main organic acids involved in the tricarboxylic acid cycle (TCA cycle), were evaluated using HPLC in the fermented and unfermented lemon juice. In addition, lactic acid and oxalic acid were also detected. After fermentation, a significant decline was found in the three organic acids, contributing to the significant decline in total titratable acidity.

Limonin and other similar compounds were the main cause of bitterness in citrus fruits. In this study, limonin and nomilin were analyzed by HPLC in fermented and control lemon juice. Nomilin was not detected in the latter. During the fermentation process, the amount of limonin increased from 0.013 ± 0.000 g/L to 0.052 ± 0.002 g/L. In the control group, no significant changes in the contents of limonin occurred. The high concentrations of limonin in lemon juice led to a pronounced bitter taste.

### 2.2. Changes in Total Phenolic and Total Flavonoid Contents

As shown in Figure 1A, the total phenolic contents of fermented and control lemon juice decreased from 1.13 ± 0.00 g/L to 0.95 ± 0.01 g/L and from 1.13 ± 0.00 g/L to 0.84 ± 0.02 g/L, respectively. Both the fermented and control groups showed decreasing trends; however, significant differences in total phenolic contents were observed between fermented and control lemon juice at 48 and 60 h (see Figure 1A).

The total flavonoid contents of fermented and control lemon juice decreased from 0.72 ± 0.01 g/L to 0.47 ± 0.01 g/L and 0.72 ± 0.01 g/L to 0.39 ± 0.00 g/L, respectively, as shown in Figure 1B. Significant differences in total flavonoid contents were observed between fermented and control lemon juice at 36, 48, and 60 h (see Figure 1B).

### 2.3. Antioxidant Properties of Fermented and Control Lemon Juice

Results of antioxidant capacities, including ABTS^+^ radical scavenging activity and total reductive ability of fermented and control lemon juice, are shown in Figure 2A,B, respectively. After 48 h fermentation, the ABTS^+^ radical scavenging activity of fermented lemon juice decreased firstly from 92.72 ± 0.13% to 75.43 ± 0.09%, compared with from 92.72 ± 0.13% to 74.06 ± 0.09% in control groups, then increased to 82.06 ± 1.00% and 76.91 ± 0.35% in fermented and control groups at 60 h. The changes of the total reductive ability of fermented lemon juice showed similar trends with ABTS^+^ radical scavenging activity, yet it decreased gradually in control groups. Significant differences of total reductive ability were observed between fermented and control lemon juice from 12 to 60 h.

### 2.4. Phenolic Compounds of Fermented and Control Lemon Juice

The results from the identification and quantification of phenolic compounds as estimated from the calibration data are shown in Table 2. A total of 10 phenolic acids and 10 flavonoids were identified and quantified. The phenolic acids, including ferulic acid, erucic acid, and cryptochlorogenic acid, were major phenolic acids with contents of 208.67 ± 0.16 mg/L, 157.36 ± 0.47 mg/L, and 81.86 ± 0.92 mg/L, respectively, in fresh lemon juice. Flavonoids including hesperidin, catechin, hesperetin, and epicatechin were determined to be the major flavonoids in fresh lemon juice, with contents of 85.77 ± 0.03 mg/L, 39.01 ± 0.13 mg/L, 26.94 ± 0.30 mg/L, and 26.76 ± 0.75 mg/L, respectively.

The contents of phenolic acids, such as ferulic acid, syringate, and *p*-coumaric acid, increased after 60 h fermentation in fermented lemon juice yet slightly decreased in control groups. However, the content of neochlorogenic acid decreased sharply from 17.19 ± 0.02 mg/L to 1.28 ± 0.91 mg/L during fermentation compared to that of the control group. Contents of gallic acid increased in both fermented and control groups during 60 h fermentation. Almost all kinds of flavonoids showed increasing contents after fermentation compared with control groups. Hesperidin, catechin, arbutin, and epicatechin were the major flavonoids in fermented lemon juice after 60 h fermentation, with contents of 98.49 ± 0.28 mg/L, 38.70 ± 0.00 mg/L, 38.49 ± 0.09 mg/L, and 31.40 ± 0.07 mg/L, respectively.

### 2.5. Sensory Evaluation

A brief sensory evaluation of the fermented and unfermented juices was conducted, as shown in Figure 3. The taste, fruitiness, and overall acceptability of lemon juice in the fermented group changed significantly compared to those of the control group. The clarity of lemon juice did not change significantly during the fermentation process. The increased overall acceptability of fermented lemon juice after 48 h and 60 h might be due to the reduced citric acid contents; however, the fruitiness decreased at the same time, which was unacceptable for some of the judges.

### 2.6. Principal Component Analysis of the Properties of Fermented and Control Lemon Juice

In order to highlight the key features of the fermented and control lemon juice, principal component analysis (PCA) was performed. In the fermented lemon juice, the total variance (70.4%) was explained by the first two components, PC1 (47.6%) and PC2 (22.8%), as shown in Figure 4A. The first group (0 h and 12 h) was on the positive side of PC1 and negative side of PC2, and it was characterized by citric acid, malic acid, cryptochlorogenic acid, neochlorogenic acid, caffeic acid, oxalic acid, erucic acid, chlorogenic acid, and ABTS^+^ radical scavenging activity. The second group (24 h and 36 h) was portrayed by hesperidin, rutin, baicalin, and succinic acid. The third group (48 h and 60 h) was characterized by most bioactive compounds such as ferulic acid, syringate, arbutin, *p*-coumaric acid, epicatechin, hesperetin, gallic acid, limonin, luteolin, quercetin, catechin *p*-hydroxybenzoic acid, and lactic acid, indicating the characteristics changed from rich of organic acids to abundant of bioactive compounds.

In the control lemon juice, the total variance (66.3%) was explained by the first two components, PC1 (44.6%) and PC2 (21.7%), as shown in Figure 4B. The first group (0 h) was on the positive side of PC1 and the negative side of PC2, and it was characterized by citric acid, rutin, succinic acid, neochlorogenic acid, chlorogenic acid, oxalic acid, and *p*-hydroxybenzoic acid. The second group (12 h) was portrayed by catechin, syringate, quercetin, malic acid, total reducing power, cryptochlorogenic acid, ABTS^+^ radical scavenging activity, baicalin, erucic acid, ferulic acid, hesperetin, and hyperoside. The third group (24 h and 36 h) was characterized by hesperidin, caffeic acid, *p*-coumaric acid, lactic acid, epicatechin, and arbutin. The fourth group (48 h and 60 h) was on the negative side of PC1 and negative side PC2, and it was characterized by limonin, luteolin, and gallic acid.

## 3. Discussion

It has been shown [12] that sugar was consumed very quickly during the fermentation of lemon juice. The rate of sugar consumption was the most rapid in the first 24 h (from 0 h to 24 h; from 27.97 ± 3.28 g/L to 6.12 ± 0.41 g/L). The contents of total sugar did not change significantly from 24 h to 60 h in the control group. During fermentation, citric acid, the main organic acid, decreased from 51.46 ± 0.11 g/L to 8.09 ± 0.05 g/L with an acidity reduction rate of 84.28%; however, no significant changes in citric acid content were observed in the control group. Malic acid contents also decreased from 5.80 ± 0.07 g/L to 3.28 ± 0.18 g/L, with an acid reduction rate of 43.45% in the fermented group compared to the control group, which showed a slight decrease content to 4.96 ± 0.01 g/L. Chen et al. demonstrated the ability of *Issatchenkia terricola* WJL-G4 to degrade both citric acid and malic acids in fermented red raspberry juice [9]. It has been reported that *Candida utilis*, a yeast species, possessed a carboxylic acid transporter for citric acid uptake [13]. Similarly, in order to utilize citric acid as a carbon source, *Issatchenkia terricola* WJL-G4 might express the citric acid transporter and the basic enzymes in related metabolic pathways. The metabolic pathway of how *Issatchenkia terricola* WJL-G4 reduce citric acid is not clear now. It is necessary to investigate the expression of carboxylic acid transporters and enzymes required for citric acid degradation in *Issatchenkia terricola* WJL-G4.

At the end of fermentation, the concentrations of succinic acid were reduced compared to the control group. It has been shown [14] that high levels of succinic acid can cause a bitter and salty taste, so the reduced contents of succinic acid might contribute to the improved taste of the fermented lemon juice. The contents of lactic acid increased from 0.61 ± 0.03 g/L to 0.89 ± 0.01 g/L at the end of fermentation. Lactic acids showed no significant changes in the control group. It has been reported [15] that high levels of lactic acid could improve the taste and overall quality of red dragon fruit wine fermented with *Saccharomyces cerevisiae*.

The total phenolic contents of both the fermented and control groups were decreasing gradually. Compared to control groups, the total phenolic contents did not show significant changes from 0 h to 36 h during fermentation but showed significant changes from 48 h to 60 h. Liu et al. reported [16] that pomegranate juice showed a decreasing trend of total phenolic contents during storage. Hashemi et al. reported [17] the possible degradation of phenolic compounds by enzymes and chemical reactions during the storage of orange juice. In this study, the contents of total phenolic possibly increased after 48 h in the fermented lemon juice, which was similar to previous observations that some specific yeasts could induce phenolic biodegradation and biosorption [18].

The contents of total flavonoid in both fermented and control groups decreased during 60 h of fermentation. No significant changes of total flavonoid contents were observed from 0 h to 24 h, but significant changes were shown from 36 h to 60 h, compared to the control group. In the fermentation of kiwifruit wine [19] using non-brewing yeast, contents of total flavonoid rose in trace amounts, while no changes occurred in blueberry wine [18]. Total flavonoid contents, as well as total phenolic contents, showed an increasing trend in fermented kiwifruit juice [20] with *Lactobacillus plantarum*. In the fermentation of red raspberry juice [17] using *S. cerevisiae*, total flavonoid contents showed a decreasing trend followed by an increasing trend. Total flavonoids showed different results during the fermentation of juices and fruit wines. The total flavonoid contents in the fermented group were significantly higher than that of the control group in the period of 36 h to 60 h, and the contents of total phenolic possibly increased after 48 h in the fermented lemon juice in this study. Meanwhile, as shown in Table 2, after 48 h of fermentation, ferulic and erucic acids were significantly increased, and the contents were higher than the control group. Therefore, it was concluded that the fermentation process increased the contents of total phenolic and total flavonoid in the period of 48 h to 60 h. The results above interpreted that *Issatchenkia terricola* WJL-G4 degraded citric acid during fermentation and had positive effects on both total phenolic and total flavonoids.

The ABTS^+^ radical scavenging activities and the total reductive abilities showed a decreasing trend in the fermented and control group. However, ABTS^+^ radical scavenging activities and the total reductive abilities in the fermented group were higher than those in the control group. During 48–60 h of fermentation, ABTS^+^ radical scavenging activities showed an increasing trend from 75.43 ± 0.09% to 82.06 ± 1.00%; similarly, the total reductive abilities also showed an increasing trend from 1.11 ± 0.002 to 1.14 ± 0.001(A_700_). Therefore, it could be concluded that the antioxidant properties might be improved after 48 h of fermentation. Chen’s study [21] showed that the antioxidant properties were not reduced during fermentation. Other studies [17,22] have shown that antioxidant properties increased after fermentation of lemon juice with *L. plantarum*. In addition, studies [20,23] have shown that phenolic compounds, including total phenolic and total flavonoid, have a strong correlation with antioxidant properties. Pearson correlation matrix of physicochemical properties and antioxidant activity in fermented lemon juice is shown in Appendix A; ABTS^+^ radical scavenging activity had a strong correlation with total phenolic and total flavonoid at the 0.05 level.

Previous studies [2,24] have shown that ferulic acid, *p*-hydroxybenzoic acid, caffeic acid, *p*-coumaric acid, chlorogenic acid, and catechic acid were detected in lemons. In this study, ferulic acid was the highest phenolic acid with a concentration of 208.67 ± 0.16 mg/L. The concentrations of caffeic acid and *p*-coumaric acid were lower. The fermentation had different effects on phenolic compounds. Ferulic acid showed an increasing trend from 208.67 ± 0.16 mg/L to 231.78 ± 1.15 mg/L; *p*-hydroxybenzoic acid and erucic acid showed a decreasing (0–24 h) and then an increasing (36–60 h) trend. In the control group, the contents of chlorogenic acid, neochlorogenic acid, ferulic acid, *p*-coumaric acid, caffeic acid, and catechin showed a slightly decreasing trend; the content of neochlorogenic acid showed a large decreasing trend; the contents of *p*-hydroxybenzoic acid and erucic acid showed a decreasing trend followed by an increasing trend, and the content of *p*-coumaric acid did not show significant changes.

It was reported [25] that the free fraction of phenolics increased, whereas ester, glycoside, and ester-bound fractions decreased after heating. Moreover, there was a decrease in total phenolic acid content after heat treatment, and it might have been consumed by yeast or transferred to other compounds [26]. It has been shown that gallic acid and *p*-hydroxybenzoic acid were reduced after fermentation [27]. Clark et al. [28] found that flavanol-type phenolic compounds such as epicatechin in white wines underwent oxidation and subsequent polymerization. Erucic acid is a common cinnamic-type phenolic acid, and hydroxycinnamic acid is readily present in ester form with other substances. The biosynthetic pathway of individual phenolic acids has been identified in citrus [29]: phenolic acids can be synthesized via the shikimic acid and the phenylpropanoid pathway. Phosphoenolpyruvate from embden meyerhof parnas and erythritose-4-phosphate from the pentose phosphate pathway produces phenylalanine via the shikimic acid pathway. Phenylalanine is then deaminated by phenylalanine ammonia-lyase and transformed into cinnamic acid. Subsequent reactions could transform cinnamic acid into *p*-coumaric acid, 2-hydroxycinnamic acid under the catalysis of cinnamate 4-hydroxylase and some other related enzymes. Therefore, we hypothesized that *Issatchenkia terricola* WJL-G4 also has a similar pathway for degrading and converting phenolic compounds.

Hesperidin was identified as the most predominant flavonoid in the present study, while hesperetin, quercetin, rutin, luteolin, arbutin, erythromycin, and baicalin were also detected, similar to previous studies [16,23]. Quercetin, luteolin, and baicalin were present at relatively low levels. Flavonoids in lemon also included naringin, diosgenin, and eriocitrin, which were not detected in the present study. The reason might be the different extraction and processed methods of flavonoids in lemon juice. Hesperidin, hesperetin, and luteolin showed increased contents compared to control groups after 60 h fermentation, with contents of 98.49 ± 0.28 mg/L, 29.21 ± 1.28 mg/L, and 9.71 ± 0.15 mg/L, respectively.

Phenolic compounds are also known to be influenced by processing and storage [4]. Likewise, pasteurization may produce thermal degradation of different compounds. Nevertheless, Dhuique-Mayer et al. reported [30] that pasteurization did not modify hesperidin content. Therefore, possible loss of phenolic compounds during sample preparation and processing might occur; thus, total phenolics and total flavonoids showed a decreasing trend in the control lemon juice, which might be due to the storage process [31].

We comprehensively analyzed the taste, fruity, and overall acceptability of the fermented juice and finally determined that the most suitable fermentation time was 48 h. Comparing the fermentation group (A) and the control group (B) in the principal component analysis (PCA), it could be seen that the components changed simultaneously during fermentation. Most of the flavonoid components had a positive correlation with fermented lemon juice at 48 h and 60 h. To conclude, the fermentation with *Issatchenkia terricola* WJL-G4 not only reduced citric acid in lemon juice but also improved the bioactive components beneficial for human health.

## 4. Materials and Methods

### 4.1. Yeast Strains and Medium

The yeast strain used in this study was isolated from the fresh fruits of red raspberry; it was identified by morphological observation, physiological, and biochemical experiments and molecular biology as *Issatchenkia terricola*, named *Issatchenkia terricola* WJL-G4 (Chinese patent No. 2019113316700) and stored at China General Microbiological Culture Collection Center (CGMCC), No. 18712 [9]. Before fermentation experiments, *Issatchenkia terricola* WJL-G4 was stored at −80 °C in a 25% glycerol solution. After thawing at room temperature, the selected colonies were inoculated into ready-made basic liquid medium in 100 mL/250 mL conical flasks, incubated at 28 °C, and shaken at 120 r/min for 12 h. Four sterile glass beads were added and shaken for 10 min to make seed solution, and the viable yeast count was determined to be 5.0 × 10^6^ CFU/mL and placed in a refrigerator at 4 °C for use. Basic culture medium compositions were composed of citric acid 20 g, yeast extract 10 g, and magnesium sulfate 1 g, dissolved in 1000 mL of distilled water.

### 4.2. Lemon Juice Preparation and Fermentation Conditions

Lemons were purchased from a local wholesale market (Harbin, Heilongjiang, China). After peeling 8 kg of lemons and squeezing the juice, the lemon juice was filtered through 8 layers of gauze to get 4 L clarified lemon juice. After pasteurization, lemon juice was inoculated with a liquid volume of 30 mL/250 mL flask and an inoculation volume of 0.5 mL (5.0 × 10^6^ CFU/mL). Lemon juice with and without (control) *Issatchenkia terricola* WJL-G4 was treated under 28 °C at 120 r/m for 12, 24, 36, 48, and 60 h.

### 4.3. Measurement of Total Titratable Acidity, Total Sugar, Reducing Sugar, and pH

The total titratable acidity of lemon juice was assessed by titration with NaOH (0.1 M) to pH 8.2 and was expressed as g/L citric acid [32]. The pH value was measured using a digital pH meter (PB-10, Sartorius, Goettingen, Germany). Total sugar contents were determined by the phenol-sulfuric acid colorimetric method, as described previously [33]. Samples of 1 mL were mixed thoroughly with 0.5 mL of 6% phenol solution and 2.5 mL sulfuric acid solution; then, the absorbance was measured at 490 nm after 30 min. The final quantification was based on a glucose calibration. Reducing sugar contents were determined by the 3,5-dinitrosalicylic acid (DNS) method, as described previously [34].

### 4.4. Determination of Total Phenolic and Total Flavonoid Contents

Total phenolic contents were determined by the Folin–Ciocalteu (FC) colorimetric method, as described previously [35] with slight modification. Diluted solution (1 mL) was added into 5 mL of 10% (m/v) Folin–Ciocalteu and 4 mL of 7.5% (m/v) Na_2_CO_3_, then mixed thoroughly and reacted in the dark for an hour. Total phenolic contents were expressed as g/L FW of gallic acid equivalent (GAE). The total flavonoid contents were determined according to the method described previously [36]. Total flavonoid contents were expressed as g/L of rutin equivalents (RE).

### 4.5. Determination of Antioxidant Activities

ABTS^+^ radical scavenging activities were determined according to the method described previously [37]. The antioxidant activities were calculated as follows:(1)$$S\;=\;\frac{{(A}_{0}{-A}_{i})}{A_{0}}\times 100\%$$

**S** was the clearance (%), **A_0_** was the absorbance value of the sample group, and **A_i_** was the absorbance value of anhydrous ethanol instead of the sample.

The total reducing abilities were determined according to the method described previously [38]. Higher absorbance indicates higher total reducing power.

### 4.6. Determination of Organic Acids and Limonin

The organic acid contents were determined according to the method described previously with slight modification [39]. Supernatant of control and fermented lemon juice were harvested after centrifugation at 4000 r/min for 15 min. After being filtered through a 0.22 μm membrane, organic acids, including citric acid, malic acid, succinic acid, oxalic acid, and lactic acid, were measured using HPLC with an Agilent ZORBAX Eclipse Plus C18 column (250 mm × 4.6 mm) (Agilent Technologies Inc., Santa Clara, CA, USA). The mobile phase consisted of 0.5% KH_2_PO_4_ (pH 2.5, solvent A) and methanol (solvent B) with ratio of A/B = 97:3 (*v/v*). The flow rate of the mobile phase was 0.7 mL/min. Volume injection was 10 µL, and the column temperature was 35 °C. The wavelength for the detection was 210 nm. The elution time was 10 min.

The limonin contents were determined according to the method described previously, with slight modification [40]. The supernatant of control and fermented lemon juice were harvested after centrifugation at 4000 r/min for 15 min. After being filtered through a 0.22 μm membrane, limonin and nomilin were measured using HPLC. The mobile phase consisted of deionized water (solvent A) and dacetonitrile (solvent B) with the ratio of A/B = 50:50 (*v/v*). The flow rate of the mobile phase was 0.2 mL/min. Volume injection was 2 µL, and the column temperature was 30 °C. The wavelength for the detection was 215 nm. The elution time was 25 min.

### 4.7. Determination of Phenolic Compounds

The phenolic compounds were determined according to the method described previously with slight modification [41]. The supernatant of fermented and control Lemon juice was harvested after centrifugation at 4000 r/min for 15 min. After being filtered through a 0.22 μm membrane, phenolic compounds in the samples were detected and separated with HPLC using an Agilent ZORBAX Eclipse Plus C18 column (250 mm × 4.6 mm). The mobile phase consisted of methanol (solvent A) and 0.02% formic acid in water (solvent B). The flow rate of the mobile phase was 0.8 mL/min. Volume injection was 10 µL, and the column temperature was 35 °C. The wavelength was 280 nm.

The elution was conducted as follows: 0~5 min, programed from 0% A to 10% A; 5~10 min, from 10% A to 20% A; 10~20 min, from 20% A to 35% A; 20~35 min, from 35% A to 40% A; 35~40 min, from 40% A to 75% A; 40~45 min, from 75% A to 10% A. Identification of the compounds was done by comparison of their retention time with those of the standards. Stock solutions of standards were prepared in methanol–dimethyl sulfoxide (DMSO) (90:10, *v/v*).

### 4.8. Sensory Analysis

Pasteurization was used before sensory evaluation. In total, 11 lemon juice (5 fermented—12, 24, 36, 48, and 60 h after fermentation; 6 unfermented—0, 12, 24, 36, 48, and 60 h of the control group) were sensory evaluated by ten untrained tasters selected from students and teaching staff of the department. The tasters were given an evaluation form with details of the grade shown in Appendix A. Grades (0–20/30, where 0 is minimal and 20/30 is maximal) were carried out based on the following sensory parameters: the samples were evaluated by clarity (organization status, sediment, turbidity, and overall sensation), flavor (purity, typicality, intensity, and overall sensation), taste (acidity, sweetness, bitter-sweet taste, harmonious taste, and overall sensation), and the overall acceptance. The sensory evaluation test was carried out in a sensory panel room. Each taster assessed 10 mL of juice, which was served in 50 mL plastic cups at room temperature; fresh water was used for palate cleansing.

### 4.9. Statistical Analysis

All samples were conducted in triplicate, and the results were expressed in means ± SD. All experimental data were analyzed using SPSS statistics software (V.26.0, SPSS Inc., Chicago, IL, USA), OriginPro software (2021, Southampton, MA, USA). One-way analysis of variance (ANOVA) followed by Duncan tests conducted by using SPSS was performed to determine the significance level at *p* < 0.05. PCA analysis and radar map were implemented by using OriginPro.

## 5. Conclusions

In this study, biological deacidification of lemon juices was performed using *Issatchenkia terricola* WJL-G4 fermentation. The results provided information on the kinetics of changes in the content of organic acids, phenolic compounds, and antioxidant activities during 60 h fermentation. At the end of fermentation, the citric acid contents of lemon juice decreased from 51.46 ± 0.11 g/L to 8.09 ± 0.05 g/L, with an acid reduction rate of 84.28%. After 48 h of fermentation, the total phenolic and total flavonoid, as well as antioxidant properties of fermented lemon juice, showed an increasing trend compared to the control group. Therefore, it could be concluded that fermentation with *Issatchenkia terricola* WJL-G4 improved the active substances and antioxidant properties of lemon juice. The organoleptic evaluation and PCA analysis showed that the taste and overall acceptability of lemon juice and the amount of bioactive substances of 48 h fermentation was the most suitable time point of fermentation.

In conclusion, the fermentation process *Issatchenkia terricola* WJL-G4 could be considered a promising method of reducing citric acid in lemon juices. The reduction of excessive amounts of citric acid helps to improve the sensory attractiveness and contributes to the development of novel value-added fermented lemon products. In the future, it would be valuable to investigate other biologically active compounds, sensory analysis, shelf-life tests, and broad assessment of the health-promoting effects of fermented lemon juices. On the other hand, the mechanism of how *Issatchenkia terricola* WJL-G4 degrades citric acid should be elucidated.

## Figures and Tables

**Figure 1 molecules-26-06712-f001:**
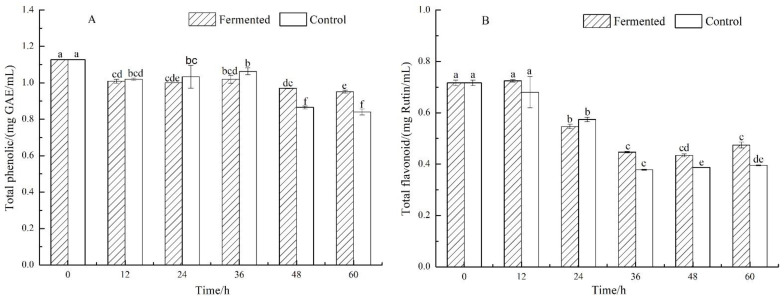
Total phenolic (**A**) and total flavonoid (**B**) contents of fermented and control lemon juice. Bars marked with different superscript lowercase letters (^a–f^) within the different periods (0 h–60 h) indicate the significant difference (*p* < 0.05) between fermented and control.

**Figure 2 molecules-26-06712-f002:**
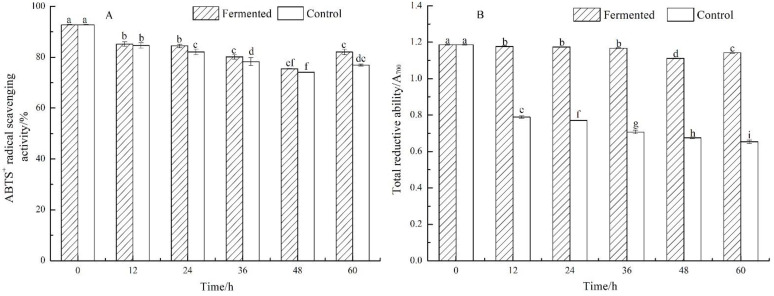
ABTS^+^ radical scavenging activity (**A**) and total reductive ability (**B**) of fermented and control lemon juice. Bars marked with different superscript lowercase letters (^a–i^) within the different periods (0 h–60 h) indicate the significant difference (*p* < 0.05) between fermented and control.

**Figure 3 molecules-26-06712-f003:**
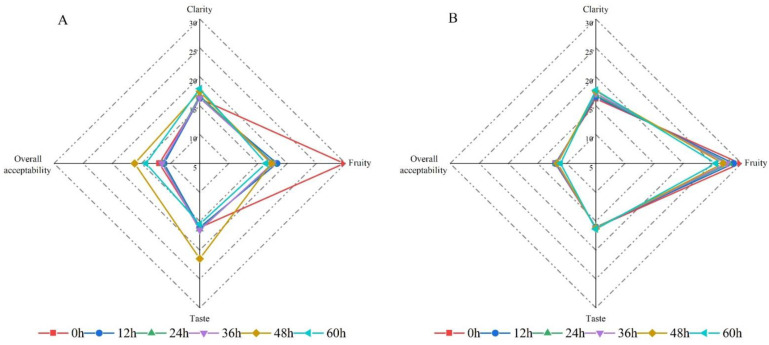
Radar plot showing sensory evaluation of fermented (**A**) and control (**B**) lemon juice.

**Figure 4 molecules-26-06712-f004:**
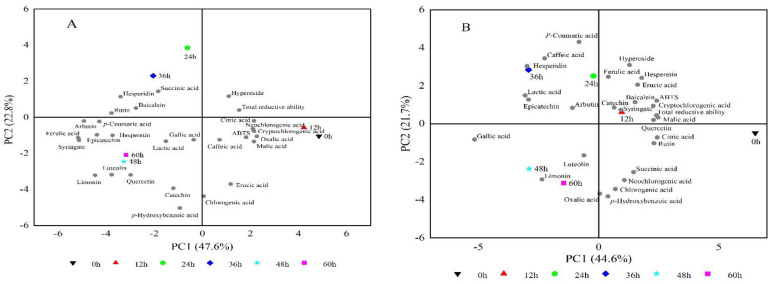
Principal component analysis (PCA) of the properties of fermented (**A**) and control (**B**) lemon juice.

**Table 1 molecules-26-06712-t001:** Physicochemical properties and organic acids of fermented and control lemon juice.

Indicators Compounds		Contents (g/L)					
		0 h	12 h	24 h	36 h	48 h	60 h
Total sugar	Fermented	27.97 ± 3.28 ^a^	17.18 ± 0.41 ^d^	6.12 ± 0.41 ^e^	6.41 ± 0.41 ^e^	6.56 ± 0.21 ^e^	6.56 ± 0.21 ^e^
	Control	27.97 ± 3.28 ^a^	24.49 ± 1.64 ^b^	27.39 ± 0.82 ^ab^	25.07 ± 2.46 ^ab^	20.42 ± 0.82 ^c^	21.00 ± 0.00 ^c^
Reducing sugar	Fermented	11.69 ± 0.70 ^b^	10.36 ± 0.70 ^c^	2.81 ± 0.05 ^d^	2.73 ± 0.21 ^d^	1.05 ± 0.05 ^e^	1.10 ± 0.02 ^e^
	Control	11.69 ± 0.70 ^b^	11.69 ± 0.70 ^b^	13.01 ± 0.70 ^a^	12.87 ± 0.47 ^ab^	14.02 ± 0.70 ^a^	13.18 ± 0.94 ^a^
Total titratable acidity	Fermented	58.59 ± 0.39 ^a^	51.09 ± 0.07 ^b^	41.38 ± 0.47 ^c^	30.80 ± 0.30 ^d^	15.30 ± 0.27 ^e^	12.20 ± 0.04 ^f^
	Control	58.60 ± 0.39 ^a^	58.60 ± 0.39 ^a^	58.57 ± 0.39 ^a^	58.54 ± 0.35 ^a^	58.58 ± 0.41 ^a^	58.53 ± 0.48 ^a^
pH	Fermented	2.61 ± 0.01 ^h^	2.67 ± 0.01 ^g^	2.78 ± 0.02 ^d^	3.02 ± 0.01 ^c^	3.68 ± 0.04 ^b^	4.31 ± 0.01 ^a^
	Control	2.61 ± 0.01 ^h^	2.62 ± 0.01 ^h^	2.71 ± 0.01 ^ef^	2.70 ± 0.01 ^efg^	2.68 ± 0.01 ^fg^	2.73 ± 0.01 ^e^
Citric acid	Fermented	51.46 ± 0.11 ^a^	45.91 ± 0.57 ^c^	37.83 ± 0.47 ^d^	25.87 ± 0.58 ^e^	9.40 ± 0.07 ^f^	8.09 ± 0.05 ^g^
	Control	51.46 ± 0.11 ^a^	51.33 ± 0.09 ^a^	50.86 ± 0.98 ^ab^	50.21 ± 0.23 ^b^	51.20 ± 0.12 ^ab^	51.09 ± 0.14 ^ab^
Malic acid	Fermented	5.80 ± 0.07 ^a^	4.77 ± 0.09 ^cd^	4.45 ± 0.04 ^d^	3.21 ± 0.04 ^e^	3.40 ± 0.42 ^e^	3.28 ± 0.18 ^e^
	Control	5.80 ± 0.07 ^a^	5.32 ± 0.18 ^b^	5.19 ± 0.04 ^b^	4.98 ± 0.16 ^bc^	5.03 ± 0.13 ^bc^	4.96 ± 0.010 ^bc^
Oxalic acid	Fermented	1.03 ± 0.01 ^ab^	0.86 ± 0.06 ^b^	0.63 ± 0.01 ^c^	0.66 ± 0.09 ^c^	0.44 ± 0.01 ^d^	0.44 ± 0.01 ^d^
	Control	1.03 ± 0.01 ^ab^	1.07 ± 0.01 ^a^	0.92 ± 0.21 ^ab^	0.89 ± 0.11 ^ab^	1.02 ± 0.02 ^ab^	1.05 ± 0.06 ^ab^
Lactic acid	Fermented	0.61 ± 0.03 ^cd^	0.33 ± 0.08 ^e^	0.51 ± 0.01 ^d^	0.50 ± 0.01 ^d^	0.10 ± 0.00 ^f^	0.89 ± 0.01 ^a^
	Control	0.61 ± 0.03 ^cd^	0.79 ± 0.15 ^ab^	0.69 ± 0.01 ^bc^	0.71 ± 0.09 ^bc^	0.69 ± 0.05 ^bc^	0.67 ± 0.08 ^bc^
Succinic acid	Fermented	1.19 ± 0.06 ^c^	0.78 ± 0.03 ^def^	1.77 ± 0.04 ^b^	2.23 ± 0.14 ^a^	0.95 ± 0.01 ^cde^	0.50 ± 0.01 ^f^
	Control	1.19 ± 0.06 ^a^	1.04 ± 0.02 ^cd^	0.63 ± 0.03 ^ef^	0.69 ± 0.06 ^def^	0.78 ± 0.05 ^def^	1.03 ± 0.45 ^cd^
Limonin	Fermented	0.013 ± 0.00 ^de^	0.018 ± 0.00 ^de^	0.018 ± 0.001 ^d^	0.028 ± 0.001 ^c^	0.037 ± 0.008 ^b^	0.052 ± 0.002 ^a^
	Control	0.013 ± 0.00 ^de^	0.011 ± 0.00 ^e^	0.012 ± 0.001 ^de^	0.013 ± 0.00 ^de^	0.014 ± 0.002 ^de^	0.014 ± 0.001 ^de^
Nomillin	Fermented	n.d.	n.d.	n.d.	n.d.	n.d.	n.d.
	Control	n.d.	n.d.	n.d.	n.d.	n.d.	n.d.

n.d.—not detected. Values are given as the mean ± standard deviation (*n* = 3), and values marked with different superscript lowercase letters (^a–h^) within the different periods (0–60 h) indicate each compound between the fermented and control are significantly different (*p* < 0.05).

**Table 2 molecules-26-06712-t002:** Phenolic compounds of fermented and control lemon juice.

Phenolic Compounds		Contents (mg/L)					
		0 h	12 h	24 h	36 h	48 h	60 h
Chlorogenic acid	Fermented	15.05 ± 0.01 ^ab^	15.27 ± 0.15 ^ab^	12.30 ± 0.05 ^c^	13.18 ± 1.03 ^abc^	15.57 ± 3.12 ^a^	13.68 ± 0.04 ^abc^
	Control	15.05 ± 0.01 ^ab^	13.14 ± 0.03 ^abc^	13.37 ± 0.75 ^abc^	12.81 ± 0.12 ^bc^	14.48 ± 0.62 ^abc^	14.12 ± 0.07 ^abc^
Neochlorogenic acid	Fermented	17.19 ± 0.02 ^a^	14.17 ± 1.22 ^d^	12.91 ± 0.10 ^e^	1.93 ± 0.07 ^f^	1.21 ± 0.11 ^f^	1.28 ± 0.91 ^f^
	Control	17.19 ± 0.02 ^a^	15.91 ± 0.21 ^b^	15.66 ± 0.60 ^bc^	14.57 ± 0.23 ^cd^	14.86 ± 0.05 ^bcd^	15.62 ± 0.02 ^bc^
Cryptochlorogenic acid	Fermented	81.86 ± 0.92 ^a^	68.12 ± 1.04 ^b^	64.33 ± 0.05 ^b^	56.67 ± 2.70 ^c^	47.16 ± 1.08 ^d^	50.16 ± 1.55 ^d^
	Control	81.86 ± 0.92 ^a^	66.94 ± 0.79 ^b^	58.53 ± 7.76 ^c^	28.15 ± 0.44 ^e^	29.29 ± 1.04 ^e^	33.81 ± 0.87 ^e^
Ferulic acid	Fermented	208.67 ± 0.16 ^c^	207.39 ± 0.06 ^c^	227.06 ± 1.68 ^b^	226.95 ± 0.11 ^b^	229.59 ± 0.39 ^ab^	231.78 ± 1.15 ^a^
	Control	208.67 ± 0.16 ^c^	197.00 ± 3.80 ^e^	204.65 ± 3.53 ^cd^	207.48 ± 1.39 ^c^	189.00 ± 3.15 ^f^	202.11 ± 0.49 ^d^
Syringate	Fermented	5.01 ± 0.01 ^cd^	4.88 ± 0.12 ^e^	5.50 ± 0.03 ^b^	5.84 ± 0.03 ^a^	5.74 ± 0.08 ^a^	5.84 ± 0.03 ^a^
	Control	5.01 ± 0.01 ^cd^	5.02 ± 0.04 ^c^	5.03 ± 0.04 ^c^	4.90 ± 0.01 ^de^	4.85 ± 0.01 ^e^	5.00 ± 0.00 ^cd^
*p*-Coumaric acid	Fermented	19.16 ± 0.01 ^d^	18.55 ± 0.12 ^e^	20.78 ± 0.03 ^b^	20.82 ± 0.01 ^b^	19.54 ± 0.14 ^c^	21.33 ± 0.07 ^a^
	Control	19.16 ± 0.01 ^d^	19.11 ± 0.19 ^d^	19.11 ± 0.10 ^d^	19.64 ± 0.39 ^c^	18.60 ± 0.10 ^e^	18.58 ± 0.09 ^e^
Gallic acid	Fermented	12.66 ± 0.10 ^f^	42.88 ± 1.42 ^a^	23.78 ± 0.79 ^d^	21.59 ± 0.37 ^de^	21.71 ± 0.97 ^de^	19.87 ± 0.29 ^e^
	Control	12.66 ± 0.10 ^f^	22.57 ± 0.73 ^d^	23.90 ± 2.81 ^d^	29.55 ± 0.60 ^bc^	28.87 ± 0.61 ^c^	31.73 ± 1.01 ^b^
Caffeic acid	Fermented	1.95 ± 0.02 ^abc^	1.95 ± 0.13 ^abc^	1.82 ± 0.11 ^cd^	1.87 ± 0.02 ^bc^	1.62 ± 0.05 ^d^	1.96 ± 0.03 ^abc^
	Control	1.95 ± 0.02 ^abc^	2.16 ± 0.01 ^a^	2.19 ± 0.02 ^a^	2.09 ± 0.04 ^ab^	2.08 ± 0.01 ^ab^	1.86 ± 0.26 ^bc^
*p*-Hydroxybenzoic acid	Fermented	14.35 ± 0.13 ^ab^	13.25 ± 0.94 ^b^	5.83 ± 0.02 ^ef^	8.29 ± 0.47 ^cd^	15.30 ± 0.37 ^a^	14.73 ± 0.05 ^ab^
	Control	14.35 ± 0.13 ^ab^	7.23 ± 0.07 ^de^	5.37 ± 1.64 ^f^	6.35 ± 0.06 ^ef^	9.58 ± 0.40 ^c^	14.46 ± 0.15 ^c^
Erucic acid	Fermented	157.36 ± 0.47 ^a^	146.61 ± 5.50 ^ab^	127.42 ± 2.59 ^cd^	127.28 ± 0.31 ^cd^	133.26 ± 1.65 ^bcd^	145.76 ± 0.17 ^ab^
	Control	157.36 ± 0.47 ^a^	137.53 ± 15.42 ^bc^	141.37 ± 2.61 ^bc^	144.04 ± 1.50 ^ab^	121.34 ± 1.20 ^d^	136.23 ± 13.31 ^bcd^
Catechin	Fermented	39.01 ± 0.13 ^a^	34.95 ± 0.45 ^bc^	22.56 ± 0.16 ^e^	38.35 ± 1.87 ^a^	35.75 ± 0.45 ^b^	38.70 ± 0.00 ^a^
	Control	39.01 ± 0.13 ^a^	33.23 ± 0.04 ^c^	29.87 ± 2.24 ^d^	30.83 ± 0.20 ^d^	35.75 ± 0.49 ^b^	21.33 ± 0.52 ^e^
Epicatechin	Fermented	26.76 ± 0.75 ^h^	29.39 ± 0.08 ^d^	29.94 ± 0.00 ^cd^	32.27 ± 0.04 ^a^	30.42 ± 0.69 ^c^	31.40 ± 0.07 ^b^
	Control	26.76 ± 0.75 ^h^	28.87 ± 0.14 ^ef^	28.42 ± 0.49 ^e^	27.38 ± 0.43 ^gh^	27.44 ± 0.06 ^fgh^	27.80 ± 0.29 ^efg^
Rutin	Fermented	13.96 ± 0.05 ^d^	13.88 ± 0.19 ^d^	23.12 ± 0.60 ^a^	16.43 ± 0.02 ^c^	18.13 ± 0.20 ^b^	18.87 ± 0.28 ^b^
	Control	13.96 ± 0.05 ^d^	13.49 ± 0.35 ^de^	12.83 ± 0.18 ^f^	12.42 ± 0.08 ^f^	12.77 ± 0.40 ^f^	12.92 ± 0.18 ^ef^
Arbutin	Fermented	17.73 ± 0.38 ^c^	16.74 ± 3.01 ^c^	44.07 ± 0.54 ^a^	46.53 ± 0.06 ^a^	43.26 ± 4.14 ^ab^	38.49 ± 0.09 ^b^
	Control	17.73 ± 0.38 ^c^	13.51 ± 0.04 ^c^	17.44 ± 3.97 ^c^	17.22 ± 2.14 ^c^	17.20 ± 3.52 ^c^	15.50 ± 0.85 ^c^
Hesperidin	Fermented	85.77 ± 0.03 ^fg^	92.30 ± 0.02 ^efg^	159.28 ± 0.37 ^b^	194.74 ± 0.85 ^a^	125.98 ± 0.94 ^d^	98.49 ± 0.28 ^e^
	Control	85.77 ± 0.03 ^fg^	54.08 ± 4.60 ^h^	135.76 ± 9.80 ^c^	166.61 ± 4.51 ^b^	84.38 ± 5.76 ^g^	94.91 ± 2.78 ^ef^
Hesperetin	Fermented	26.94 ± 0.30 ^bcd^	29.37 ± 0.16 ^abc^	32.18 ± 2.81 ^ab^	29.28 ± 0.76 ^abc^	33.20 ± 1.14 ^a^	29.21 ± 1.28 ^abc^
	Control	26.94 ± 0.30 ^bcd^	24.69 ± 3.60 ^cd^	23.56 ± 3.06 ^cd^	24.42 ± 1.73 ^cd^	21.39 ± 3.90 ^d^	21.59 ± 3.60 ^d^
Naringin	Fermented	n.d.	n.d.	n.d.	n.d.	n.d.	n.d.
	Control	n.d.	n.d.	n.d.	n.d.	n.d.	n.d.
Hyperoside	Fermented	13.15 ± 0.07 ^ab^	13.58 ± 0.76 ^ab^	14.06 ± 0.29 ^a^	12.99 ± 0.65 ^ab^	9.97 ± 1.65 ^ab^	10.46 ± 0.18 ^ab^
	Control	13.15 ± 0.07 ^ab^	11.98 ± 1.73 ^ab^	13.50 ± 3.34 ^ab^	11.33 ± 0.68 ^ab^	9.38 ± 0.74 ^b^	10.44 ± 0.92 ^ab^
Quercetin	Fermented	6.10 ± 0.39 ^a^	3.76 ± 0.10 ^c^	4.96 ± 0.32 ^b^	5.01 ± 0.10 ^b^	6.12 ± 0.24 ^a^	6.16 ± 0.28 ^a^
	Control	6.10 ± 0.39 ^a^	4.95 ± 0.12 ^b^	4.87 ± 0.11 ^b^	4.91 ± 0.27 ^b^	4.72 ± 0.15 ^b^	4.76 ± 0.18 ^b^
Luteolin	Fermented	4.27 ± 0.04 ^cd^	4.11 ± 0.05 ^cd^	4.69 ± 0.84 ^bcd^	4.63 ± 1.52 ^bcd^	5.85 ± 0.03 ^b^	9.71 ± 0.15 ^a^
	Control	4.27 ± 0.04 ^cd^	3.78 ± 0.07 ^cd^	4.71 ± 0.10 ^bcd^	4.27 ± 0.00 ^d^	3.83 ± 0.07 ^cd^	4.92 ± 0.02 ^bc^
Baicalein	Fermented	3.87 ± 0.28 ^ab^	4.16 ± 0.11 ^a^	4.43 ± 0.25 ^a^	4.54 ± 0.04 ^a^	4.02 ± 0.00 ^ab^	4.53 ± 0.10 ^a^
	Control	3.87 ± 0.28 ^ab^	3.92 ± 0.61 ^ab^	4.01 ± 0.32 ^ab^	2.31 ± 0.02 ^c^	3.16 ± 0.84 ^bc^	2.45 ± 0.64 ^c^

n.d.—not detected. Values are given as the mean ± standard deviation (*n* = 3), and values marked with different superscript lowercase letters (^a–h^) within the different periods (0 h–60 h) indicate each compound between the fermented and control are significantly different (*p* < 0.05).

## Data Availability

The study did not report any data.

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
