# Peer review of "Changes in Organic Acids, Phenolic Compounds, and Antioxidant Activities of Lemon Juice Fermented by Issatchenkia terricola"

_molecules, 2021, doi:10.3390/molecules26216712_

Round 1
Reviewer 1 Report
The manuscript demonstrated that fermentation of lemon juice with Issatchenkia terricola WJL-G4 could be considered as a promising method of reducing citric acid, while simultaneously increasing levels of health-promoting phenolic compounds, and enhancing antioxidant activities in lemon juices. The research is meaningful and the results should be of interest to the journal readers.
Still, some points should be revised in the manuscript.
Lines 29 – 32: Please, add the literature where you found data about lemon cultivation in China.
Line 182. The genus name must be given in full at least the first time when is mentioned in the manuscript.
Line 200: Please, correct “thatpomegranate”
Lines 213 and 232: Now you can abbreviated the genus names (S. cerevisiae, L. plantarum).
Line 229: Please explain what (A700) means.
Line 231. Double brackets [[17,22]]. Please, correct.
Line 282: My suggestion is to add some discussion about sensory and PCA analyses in the discussion part.
Author Response
Dear Editor-in-Chief in Molecules:
Thank you very much for your help in processing the review of our manuscript (Manuscript ID molecules-1443860). We have carefully read the thoughtful comments from you and reviewers and found that these suggestions are helpful for us to improve our manuscript. On the basis of the enlightening questions and helpful advices, we have now completed the revision of our manuscript. The itemized responses to the reviewers’ comments are listed in the succeeding sheets. We hope that all these corrections and revisions would be satisfactory. Thanks a lot, again.
- Title: Changes in organic acids, phenolic compounds and antioxidant activities of lemon juice fermented by Issatchenkia terricola
- Manuscript type: Article
- Corresponding author: Jinling Wang
- Full author names: Biao Liu , Dongxia Yuan , Qiaoyue Li , Xin Zhou , Hao Wu , Yihong Bao , Hongyun Lu , Ting Luo
Sincerely,
M.S. Jinling Wang
School of Forestry,
Northeast Forestry University,
Harbin, Heilongjiang, 150040, P R China.
2021-10-31
Responses to comments of Editor
Thank you for your serious and constructive comments on our manuscript. According to your suggestion, the manuscript has been revised as a letter to editor. The revisions we have made are as follows:
Ø 1 Lines 29 – 32: Please, add the literature where you found data about lemon cultivation in China.
Reply:
Thank you for your constructive and helpful suggestion. We have modified the article according to your request.
Ø 2 Line 182. The genus name must be given in full at least the first time when is mentioned in the manuscript.
Reply:
Thank you for your constructive and helpful suggestion. We have modified the article according to your request.
Ø 3 Line 200: Please, correct “thatpomegranate”
Reply:
Thank you for your constructive and helpful suggestion. We have modified the article according to your request.
Ø 4 Lines 213 and 232: Now you can abbreviated the genus names (S. cerevisiae, L. plantarum).
Reply:
Thank you for your constructive and helpful suggestion. We have modified the article according to your request.
Ø 5 Line 229: Please explain what (A700) means.
Reply:
Thank you for your constructive and helpful suggestion. We have modified the article according to your request.
A700 indicates the absorbance at 700nm
Ø 6 Line 231. Double brackets [[17,22]]. Please, correct
Reply:
Thank you for your constructive and helpful suggestion. We have modified the article according to your request.
Ø 7 Line 282: My suggestion is to add some discussion about sensory and PCA analyses in the discussion part.
Reply:
Thank you for your constructive and helpful suggestion. We have modified the article according to your request.
We comprehensively analyzed the taste, fruity and overall acceptability of the fermented juice, and finally determined that the most suitable fermentation time was 48 h. Comparing the fermentation group (A) and the control group (B) in the principal component analysis (PCA), it could be seen that the components changed simultaneously during fermentation. Most of the flavonoid components had positive correlation with fermented lemon juice at 48 h and 60 h. To conclude, the fermentation with Issatchenkia terricola WJL-G4 not only reduced citric acid in lemon juice, but also improved the bioactive components benefited for human health.

Reviewer 2 Report
This paper presents a biological deacidification of lemon juice by means of a specific mediator Issatchenkia terricola. The system is studied for the first time thus making the research original and interesting. The main conclusion is that by this fermentation the juice composition is shifted from rich of organic acids to rich of bioactive components with improved antioxidant activity.
The manuscript is well arranged and clearly presented. The conclusions are supported by the results. I have found only one contradiction. At a number of places (rows 203,217,221,389) it is clamed that the fermentation increases the total phenolic contents, while Fig. 1 A does not show such a result even after 48 hours.
Some small corrections are shown below:
row 72-73 -
(58.59 72± 0.39 to 12.20 ± 0.04 g/L). The variation of pH ranged from 2.61 ± 0.01 to 4.31 ± 0.01.
Repetition. This data is just given abovе. It will be better to replace by (see above and Table 1).
r.75,76 - detected in fermented and control lemon
juice using HPLC, where nomilin was not detected.
To become: analyzed by HPLC in fermented and control lemon
juice. Nomilin was not detected in the latter.
r.88. at 48 and 60 h - add (see Fig. 1 A).
r.88. The total flavonoid contents - New paragraph
Tables and Figures: explain the meaning of a,b,c,d...
r. 117 - Contents should become The contents
r. 168 - benifited should become benefited
r. 191 - tasty should become taste
r. 200 - thatpomegranate shoud become that pomegranate
r. 204 - with previous researches should become to previous observations
r. 250 - It was reported that[25] should become It was reported [25] that
r. 266 - flavonoids should become flavonoid
Paragraph r. 250-265. Show more clearly the relation to the present work.
r. 268 - studies[16,23], while quercetin, should become studies[16,23]. Quercetin, ...
r. 271 - flavonoid should become flavonoids
r. 273 - compared with control should become compared to
r. 351 - After filtered should become After being filtered
r. 398 - help to improve the sensory attractiveness and contribute
should become helps to improve the sensory attractiveness and contributes
My opinion is that the manuscript should be accepted for publication. Minor corrections are needed.
Author Response
Dear Editor-in-Chief in Molecules:
Thank you very much for your help in processing the review of our manuscript (Manuscript ID molecules-1443860). We have carefully read the thoughtful comments from you and reviewers and found that these suggestions are helpful for us to improve our manuscript. On the basis of the enlightening questions and helpful advices, we have now completed the revision of our manuscript. The itemized responses to the reviewers’ comments are listed in the succeeding sheets. We hope that all these corrections and revisions would be satisfactory. Thanks a lot, again.
- Title: Changes in organic acids, phenolic compounds and antioxidant activities of lemon juice fermented by Issatchenkia terricola
- Manuscript type: Article
- Corresponding author: Jinling Wang
- Full author names: Biao Liu , Dongxia Yuan , Qiaoyue Li , Xin Zhou , Hao Wu , Yihong Bao , Hongyun Lu , Ting Luo
Sincerely,
M.S. Jinling Wang
School of Forestry,
Northeast Forestry University,
Harbin, Heilongjiang, 150040, P R China.
2021-10-31
Responses to comments of Editor
Thank you for your serious and constructive comments on our manuscript. According to your suggestion, the manuscript has been revised as a letter to editor. The revisions we have made are as follows:
Ø 1 The manuscript is well arranged and clearly presented. The conclusions are supported by the results. I have found only one contradiction. At a number of places (rows 203,217,221,389) it is claimed that the fermentation increases the total phenolic contents, while Fig. 1 A does not show such a result even after 48 hours.
Reply:
Thank you for your constructive and helpful suggestion.
During the fermentation of lemon juice, the total phenolic of both the fermented and control groups showed decreasing trends, suggesting that the total phenolic would be naturally degraded under the same conditions and that it is difficult to explain the role of yeast in the fermentation process. However, significant differences of total phenolic contents were observed between fermented and control lemon juice at 48 and 60 h (see Fig. 1 A), the fermentation (0.97 g/L) of total phenolic was higher than the control group (0.85 g/L) at 48 h. We believed that the fermentation process with Issatchenkia terricola WJL-G4 possibly increased the contents of total Phenolic. Meanwhile, as shown in table 2, after 48 h of fermentation, ferulic and erucic acids were significantly increased, and the contents were higher than the control group. So, we concluded that the fermentation increases the total phenolic contents.
Ø 2 row 72-73 -(58.59 72± 0.39 to 12.20 ± 0.04 g/L). The variation of pH ranged from 2.61 ± 0.01 to 4.31 ± 0.01. Repetition. This data is just given abovе. It will be better to replace by (see above and Table 1).
Reply:
Thank you for your constructive and helpful suggestion. We have modified the article according to your request.
Ø 3 r.75,76 - detected in fermented and control lemon juice using HPLC, where nomilin was not detected. To become: analyzed by HPLC in fermented and control lemon juice. Nomilin was not detected in the latter.
Reply:
Thank you for your constructive and helpful suggestion. We have modified the article according to your request.
Ø 4 r.88. at 48 and 60 h - add (see Fig. 1 A).
Reply:
Thank you for your constructive and helpful suggestion. We have modified the article according to your request.
Ø 5 r.88. The total flavonoid contents - New paragraph
Reply:
Thank you for your constructive and helpful suggestion. We have modified the article according to your request.
Ø 6 Tables and Figures: explain the meaning of a,b,c,d...
Reply:
Thank you for your constructive and helpful suggestion. We have modified the article according to your request.
Figures 1 and 2: Bars marked with different superscript lowercase letters (a, b, c, d, e, f, g and h) within the different periods (0 h – 60 h) indicate the significant difference (p < 0.05) between fermented and control.
Table 1 and 2: Values marked with different superscript lowercase letters (a, b, c, d, e, f, g and h) within the different periods (0 h – 60 h) indicate the each compounds between the fermented and control are significantly different (p < 0.05).
Ø 7 r. 117 - Contents should become The contents
Reply:
Thank you for your constructive and helpful suggestion. We have modified the article according to your request.
Ø 8 r. 168 - benifited should become benefited
Reply:
Thank you for your constructive and helpful suggestion. We have modified the article according to your request.
Ø 9 r. 191 - tasty should become taste
Reply:
Thank you for your constructive and helpful suggestion. We have modified the article according to your request.
Ø 10 r. 200 - thatpomegranate shoud become that pomegranate
Reply:
Thank you for your constructive and helpful suggestion. We have modified the article according to your request.
Ø 11 r. 204 - with previous researches should become to previous observations
Reply:
Thank you for your constructive and helpful suggestion. We have modified the article according to your request.
Ø 12 r. 250 - It was reported that[25] should become It was reported [25] that
Reply:
Thank you for your constructive and helpful suggestion. We have modified the article according to your request.
Ø 13 r. 266 - flavonoids should become flavonoid
Reply:
Thank you for your constructive and helpful suggestion. We have modified the article according to your request.
Ø 14 Paragraph r. 250-265. Show more clearly the relation to the present work.
Reply:
Thank you for your constructive and helpful suggestion. We have modified the article according to your request.
Therefore, we hypothesized that Issatchenkia terricola WJL-G4 also has a similar pathway for degrading and converting phenolic compounds.
Ø 15 r. 268 - studies[16,23], while quercetin, should become studies[16,23]. Quercetin,
Reply:
Thank you for your constructive and helpful suggestion. We have modified the article according to your request.
Ø 16 r. 271 - flavonoid should become flavonoids
Reply:
Thank you for your constructive and helpful suggestion. We have modified the article according to your request.
Ø 17 r. 273 - compared with control should become compared to
Reply:
Thank you for your constructive and helpful suggestion. We have modified the article according to your request.
Ø 18 r. 351 - After filtered should become After being filtered
Reply:
Thank you for your constructive and helpful suggestion. We have modified the article according to your request.
Ø 19 r. 398 - help to improve the sensory attractiveness and contribute should become helps to improve the sensory attractiveness and contributes
Reply:
Thank you for your constructive and helpful suggestion. We have modified the article according to your request.
